# Explore Data Left Behind in Reinforcement Learning for Reasoning Language Models

## Abstract

Reinforcement Learning with Verifiable Rewards (RLVR) has emerged as an effective approach for improving the reasoning abilities of large language models (LLMs). The Group Relative Policy Optimization (GRPO) family has demonstrated strong performance in training LLMs with RLVR. However, as models train longer and scale larger, more training prompts become residual prompts—those with zero-variance rewards that provide no training signal. Consequently, fewer prompts contribute to training, reducing diversity and hindering effectiveness. To fully exploit these residual prompts, we propose the **E**xplore **R**esidual Prompts in **P**olicy **O**ptimization (ERPO) framework, which encourages exploration on residual prompts and reactivates their training signals. ERPO maintains a history tracker for each prompt and adaptively increases the sampling temperature for residual prompts that previously produced all-correct responses. This encourages the model to generate more diverse reasoning traces, introducing incorrect responses that revive training signals. Empirical results on the Qwen2.5 series demonstrate that ERPO consistently surpasses strong baselines across multiple mathematical reasoning benchmarks.

## 1 Introduction

Large language models (LLMs) have become the foundation of modern artificial intelligence, exhibiting strong performance across domains such as mathematics, programming, and scientific problem solving (Team et al., 2023; Guo et al., 2025; Yang et al., 2025a). A central factor behind these advancements is their capacity for extended reasoning, where models construct coherent, multi-step chains of thought to address complex tasks (Wei et al., 2022; Yao et al., 2023; Muennighoff et al., 2025). Reinforcement learning (RL) has emerged as a key paradigm for strengthening this capability, enabling LLMs to refine their responses through interaction-driven feedback and alignment with verifiable signals or human preferences (Schulman et al., 2017; Ouyang et al., 2022; Rafailov et al., 2023). In particular, reinforcement learning with verifiable rewards (RLVR) has proven especially effective, as it leverages tasks with automatically checkable outcomes to provide reliable supervision for scaling reasoning abilities (Shao et al., 2024; Guo et al., 2025; Yang et al., 2025a).

Among recent advances in reinforcement learning for LLMs, Group Relative Policy Optimization (GRPO) has emerged as a widely adopted RLVR framework (Shao et al., 2024; Guo et al., 2025). Building on this foundation, subsequent research has sought to address key issues of GRPO, including entropy collapse, reward noise, and training instability (Yu et al., 2025; Cui et al., 2025; Zheng et al., 2025a). Furthermore, as an on-policy algorithm, GRPO has motivated efforts to develop more effective sampling strategies beyond basic random decoding (Xu et al., 2025; Zheng et al., 2025c; Hou et al., 2025).

In this work, we identify a limitation shared by the GRPO family of algorithms: as training steps and model size increase, more training prompts become residual prompts that no longer provide training signals yet still contain valuable information that can benefit model performance. Residual prompts are those that initially provide effective training signals at the beginning of training but eventually provide zero training signal or are filtered out by the RL algorithms as the well-trained policy generates all-correct responses for them. This reduces training diversity over time and ultimately hinders further improvement through RL. Furthermore, residual prompts retain learning potential that can be leveraged to further improve model performance, as they help the model retain acquired abilities

Table 1: Proportion of prompts with all-correct responses under different sampling temperatures and model scales. The proportion increases with RL training process and larger model sizes, leaving more residual prompts, thereby reducing diversity and wasting valuable training signals.

|  | $T = 1.0$ | $T = 1.1$ | $T = 1.2$ |
|---|---|---|---|
| Qwen2.5-3B | 0% | – | – |
| Qwen2.5-3B + DAPO | 8.7% | 6.2% | 2.8% |
| Qwen2.5-7B | 0% | – | – |
| Qwen2.5-7B + DAPO | 21.3% | 15.5% | 5.5% |
| Qwen2.5-32B | 0% | – | – |
| Qwen2.5-32B + DAPO | 74.8% | 62.1 | 34.8% |

and may yield novel reasoning traces. Moreover, residual prompts are not necessarily robust—small perturbations, such as increasing the sampling temperature, can easily induce errors. Table 1 reports the proportion of residual prompts with all-correct responses in the training data under different sampling temperatures and model scales.

To better exploit the residual prompts left behind during training, we propose the **E**xplore **R**esidual Prompts in **P**olicy **O**ptimization (ERPO) framework. ERPO introduces a novel sampling strategy that maintains a history tracker for each prompt and adaptively increases the sampling temperature for residual prompts that have previously produced all-correct responses. Specifically, ERPO records how many times a model generates all-correct responses for each prompt, and the sampling temperature is determined by this count. The more frequently a prompt yields all-correct responses, the higher the sampling temperature assigned to it, thereby encouraging greater exploration. As shown in Table 1, increasing the sampling temperature enables the model to explore more diverse reasoning traces and generate incorrect responses, which reactivates the training signal and alleviates the collapse of prompt diversity.

Overall, **our contributions** can be summarized as follows:

- We identify a key limitation of the GRPO family: residual prompts accumulate as training progresses and models scale, leading to reduced training diversity and the loss of valuable training signals from residual prompts.

- We propose the ERPO framework, which encourages models to adaptively explore residual data and recover their learning potential. ERPO maintains a history tracker for each prompt and adaptively increases the sampling temperature for residual prompts.

- Extensive experiments on several math reasoning benchmarks demonstrate the effectiveness of ERPO in both average and majority-vote evaluations, with particularly strong improvements on data that are likely not contaminated, such as AIME2025.

## 2 RELATED WORK

**Reinforcement learning for LLM reasoning.** Reinforcement learning (RL) has become a central approach for enhancing the reasoning abilities of large language models (LLMs) in domains such as mathematics, programming, and problem solving (Dubey et al., 2024; Zhou et al., 2025). Early general-purpose algorithms like Proximal Policy Optimization (PPO) provided a practical framework for fine-tuning LLMs through sampled rollouts and reward feedback (Schulman et al., 2015; 2017). More recently, RLVR methods such as Group Relative Policy Optimization (GRPO) have emerged as effective alternatives to PPO, removing the critic model while maintaining strong performance on reasoning benchmarks (Guo et al., 2025; Shao et al., 2024). Several extensions have been proposed to address the limitations of GRPO: Cui et al. (2025); Wang et al. (2025); Cheng et al. (2025); Zheng et al. (2025c) mitigates the entropy collapse problem during training; Zheng et al. (2025a); Yang et al. (2025b) aims to stabilize the optimization process, and DAPO (Yu et al., 2025) tackles both issues while filtering noisy rewards for training data. However, all these methods obtain no training signal from residual prompts, thereby missing valuable information during training. To

address this limitation, `ERPO` reactivates the training signal of residual prompts and learns useful information from them.

**Data Sampling Strategies.** The outputs of LLMs rely heavily on data sampling strategies to balance diversity and quality. Common strategies include greedy search, beam search, and various random sampling techniques such as top-k and top-p (Zhao et al., 2023; Minaee et al., 2024). In RLVR, the model generates on-policy responses and assigns them verifiable rewards during training. Basic random decoding is widely used in RLVR algorithms such as GRPO and DAPO (Guo et al., 2025; Yu et al., 2025). Beyond this, several works explore alternative sampling strategies. Hou et al. (2025) leverages tree search to find correct responses with higher probability. Zheng et al. (2025c) forks responses at high-entropy tokens. Shrivastava et al. (2025) dynamically allocates additional training resources to harder problems based on real-time difficulty estimates. Xu et al. (2025) selects a subset of responses to maximize reward variation. Zheng et al. (2025b) predicts and skips uninformative prompts using reward training dynamics. Zhang et al. (2025) progressively exposes the model to increasingly challenging samples. Nevertheless, none of these methods are specifically designed to leverage information from residual prompts.

## 3 PRELIMINARIES

**Notation** We define an autoregressive language model parameterized by $\theta$ as a policy $\pi_\theta$. Let $q$ denote a query and $\mathcal{D}$ the query set. For a response $o$ to query $q$, its likelihood under $\pi_\theta$ is expressed as

$$\pi_\theta(o \mid q) = \prod_{t=1}^{|o|} \pi_\theta(o_{i,t} \mid q, o_{i,<t}), \tag{1}$$

where $|o|$ is the number of tokens in $o$.

**Group Relative Policy Optimization (GRPO)** (Shao et al., 2024; Guo et al., 2025) has shown strong effectiveness for fine-tuning LLMs. Unlike traditional approaches that rely on a critic network of comparable size to the policy, GRPO estimates the baseline directly from group-level rewards. For a specific question-answer pair $(q, a)$, the behavior policy $\pi_{\theta_{\text{old}}}$ samples a group of $G$ individual responses $\{o_i\}_{i=1}^G$. Then, the advantage of the $i$-th response is calculated by normalizing the group-level rewards $\{R_i\}_{i=1}^G$:

$$\hat{A}_{i,t} = \frac{r_i - \text{mean}(\{R_i\}_{i=1}^G)}{\text{std}(\{R_i\}_{i=1}^G)}. \tag{2}$$

Building on the group-normalized advantages, GRPO optimizes the policy with a clipped objective that stabilizes updates and a KL regularization term that constrains divergence from the reference model. The objective is defined as:

$$\mathcal{J}_{\text{GRPO}}(\theta) = \mathbb{E}_{(q,a)\sim\mathcal{D}, \{o_i\}_{i=1}^G \sim \pi_{\theta_{\text{old}}}(\cdot|q)}$$

$$\left[ \frac{1}{G} \sum_{i=1}^G \frac{1}{|o_i|} \sum_{t=1}^{|o_i|} \left( \min\left( r_{i,t}(\theta)\hat{A}_{i,t}, \ \text{clip}\left(r_{i,t}(\theta), 1-\varepsilon, 1+\varepsilon\right)\hat{A}_{i,t} \right) \right. \tag{3}$$

$$\left. - \beta D_{\text{KL}}(\pi_\theta||\pi_{\text{ref}}) \right) \right],$$

where $r_{i,t}(\theta)$ is the importance ratio between the old and new policy:

$$r_{i,t}(\theta) = \frac{\pi_\theta(o_{i,t} \mid q, o_{i,<t})}{\pi_{\theta_{\text{old}}}(o_{i,t} \mid q, o_{i,<t})}. \tag{4}$$

**Decoupled Clip and Dynamic sAmpling Policy Optimization (DAPO)** (Yu et al., 2025) introduces four key improvements: Clip-Higher promotes output diversity and mitigates entropy collapse; Dynamic Sampling is designed to enhance training efficiency and stability; Token-Level Policy Gradient Loss plays a critical role in handling long chain-of-thought reasoning; and Overlong Reward Shaping reduces reward noise while stabilizing optimization. Building on these components, DAPO optimizes the policy with the following objective:

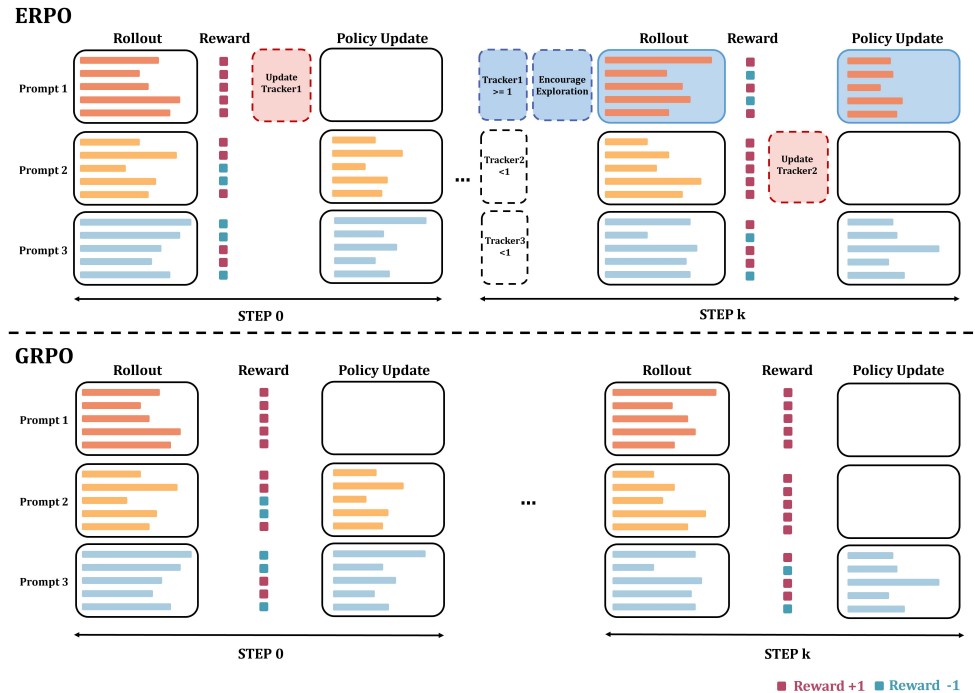

Figure 1: Comparison between ERPO and GRPO. During RL training, the policy gradually learns from the training data, resulting in more residual prompts with all-correct responses. In GRPO, residual prompts yield zero-variance rewards and thus provide no training signal for policy updates, reducing the effectiveness of training data. In contrast, ERPO maintains a tracker for each prompt to record the number of times it produces all-correct responses, and adaptively encourages exploration on residual prompts to trigger incorrect responses and reactivate the training signal.

$$
\begin{aligned}
\mathcal{J}_{\text{DAPO}}(\theta) = \quad & \mathbb{E}_{(q,a)\sim\mathcal{D},\{o_i\}_{i=1}^{G}\sim\pi_{\theta_{\text{old}}}(\cdot|q)} \\
& \left[ \frac{1}{\sum_{i=1}^{G}|o_i|} \sum_{i=1}^{G}\sum_{t=1}^{|o_i|} \min\left( r_{i,t}(\theta)\hat{A}_{i,t},\ \text{clip}\left(r_{i,t}(\theta), 1-\varepsilon_{\text{low}}, 1+\varepsilon_{\text{high}}\right)\hat{A}_{i,t}\right)\right] \quad (5) \\
\text{s.t.} \quad & 0 < \left|\{o_i \mid \texttt{is\_equivalent}(a, o_i)\}\right| < G,
\end{aligned}
$$

where $a$ is the ground-truth answer of query $q$.

## 4 METHODOLOGY

ERPO is proposed to leverage the training information contained in residual prompts to improve reinforcement learning for reasoning language models. Section 4.1 introduces the idea of reactivating the training signal from residual prompts that are otherwise discarded during RL training. Section 4.2 describes how ERPO predicts whether a prompt is residual and adaptively modifies the sampling strategy. It further explains how ERPO adjusts the sampling temperature to encourage different levels of exploration based on a history tracker.

### 4.1 REACTIVATE TRAINING SIGNAL

Current RLVR algorithms usually discard residual prompts that contain all-correct responses by assigning them zero advantage (Guo et al., 2025) or by directly filtering them out from the training batch (Yu et al., 2025). Consequently, available training prompts continually decreases with ongoing

RL training and increasing model size, leading to a gradual reduction in both the size and diversity of the training dataset. As shown in Table 1, larger models with longer training generally produce more residual prompts that yield all-correct responses. Furthermore, as training progresses and the policy evolves, new reasoning traces and directions may be generated for the residual prompts, which can help the model learn more diverse reasoning patterns. In addition, training on residual prompts can reinforce the reasoning abilities the model has already acquired.

To leverage the training signal from residual prompts that are left behind by current RL algorithms, we propose a simple method to reactivate them. For residual prompts with all-correct responses, we replace the zero advantage with a small positive advantage by introducing a pseudo-negative reward into the advantage computation. The new Reactivated Advantage (RA) for prompts with all-correct responses is:

$$\hat{RA}_{i,t} = \frac{r_i - \text{mean}\big(\{R_i^+\}_{i=1}^G \cup \{R^-\}\big)}{\text{std}\big(\{R_i^+\}_{i=1}^G \cup \{R^-\}\big)}. \tag{6}$$

where $R^+$ is the reward for correct responses and $R^-$ is the reward for incorrect responses. Using RA, residual prompts with all-correct responses still retain a small positive advantage, providing a valid training signal instead of being discarded by the RL algorithm.

## 4.2 EXPLORE RESIDUAL PROMPTS IN POLICY OPTIMIZATION

Although using the reactivated advantage can force the model to learn information from residual prompts, residual prompts will dominate the trainig along the training process and model scales up, leaving less negative feedback and impede the training effectiveness. (Chen et al., 2025). Furthermore, the model may suffer from an imbalance between exploration and exploitation, restricting exploration to a narrow search space and potentially causing overfitting on all-correct prompts, particularly at larger model scales. (Xiong et al., 2025)

To address this limitation, we propose to adaptively encourage exploration on residual prompts by controling their sampling temperature. As shown in Table 1, a higher sampling temperature can trigger incorrect responses, thereby reactivating the training signals of residual prompts. Note that training data typically exhibit strong temporal correlations across epochs (Zheng et al., 2022), meaning that a prompt producing all-correct responses in the current epoch is likely to do so again in the following epoch (Zheng et al., 2025b). Thus, we can maintain a history tracker $H_i$ to track how many times the policy generates all-correct responses for a prompt $q_i$:

$$H_i^{(0)} = 0, \qquad H_i^{(t)} = H_i^{(t-1)} + \mathbf{1}_{q_i \text{ has all-correct responses at step } t} \tag{7}$$

Then $H_i$ is used to determine whether we should assign a larger sampling temperature to prompt $q_i$. If $H_i$ is greater than 0, it means that prompt $q_i$ is already easy for the policy to generate all-correct responses, and it is very likely to provide no training signals the next time the policy samples it. Therefore, we assign a larger sampling temperature to prompts with $H_i > 0$ to encourage more exploration of their reasoning traces and to reactivate the training signal by triggering incorrect responses.

Since the robustness of prompts varies, some residual prompts require only a marginal increase in sampling temperature to induce incorrect responses, whereas others necessitate substantially larger adjustments. At the same time, it is essential to preserve the benefits of on-policy learning by constraining distributional shifts within a reasonable range to ensure stable and effective training. Assigning excessively large sampling temperatures is particularly detrimental for prompts with lower robustness. This trade-off highlights the difficulty of selecting a single, unified sampling temperature that can consistently induce incorrect responses, enhance exploration, and maintain a manageable distribution shift across all residual prompts. Therefore, ERPO introduces a prompt-adaptive adjustment of the sampling temperature:

$$T_i^{(t)} = \min(T_0 + T_s \cdot H_i^{(t)}, T_{max}) \tag{8}$$

where $T_0$, $T_{max}$, and $T_s$ are hyperparameters representing the initial temperature, maximum temperature, and temperature step size, respectively. In this way, ERPO gradually increases the sampling temperature of residual prompts until the policy generates incorrect responses. This enables ERPO to strike a balance between reactivating training signals, encouraging exploration, and maintaining a reasonable distribution shift. In general, our ERPO framework can be summarized in Algorithm 1:

---

**Algorithm 1** ERPO framework

---

**Input:** Policy $\pi_\theta$, reward model $R$,

      Prompt set $\mathcal{D} = \{q_i\}_{i=1}^N$, history tracker $\{H_i\}_{i=1}^N$ (init. $H_i^{(0)}=0$),

      Rollouts per prompt $n$, temperatures $(T_0, T_{\max})$, step size $T_s$, steps $K$

**Output:** Updated policy $\pi_{\theta_{\text{updated}}}$

 1: **for** $t = 1, 2, \ldots, K$ **do**

 2:     Sample a mini-batch $\mathbf{q} \subseteq \mathcal{D}$

 3:     **for** each $q_i \in \mathbf{q}$ **do**

 4:         $T_i^{(t)} \leftarrow \min\bigl(T_0 + T_s \cdot H_i^{(t-1)}, T_{\max}\bigr)$

 5:         Sample $n$ rollouts $\mathbf{o_i} = (o_1, \ldots, o_n)$ for $q_i$ using $\pi_\theta$ at temperature $T_i^{(t)}$

 6:         Compute rewards $\mathbf{r_i} = (r_1, \ldots, r_n)$ with $R$;  $\text{acc} \leftarrow \mathbf{1}_{\text{all } o_j \text{ correct}}$

 7:         $H_i^{(t)} \leftarrow H_i^{(t-1)} + \mathbf{1}_{\text{acc}=1}$

 8:     **end for**

 9:     Update $\pi_\theta$ using an RL algorithm with data $\mathcal{B} = \{(\mathbf{q}, \mathbf{o}, \mathbf{r})\}$

10: **end for**

11: **return** $\pi_{\theta_{\text{updated}}} \leftarrow \pi_\theta$

---

## 5 EXPERIMENTS

In this section, we first outline the implementation details, including training details and evaluation. We then present the main results, comparing ERPO against baseline approaches across several math reasoning benchmarks. Finally, we provide additional experimental results to support further analysis.

### 5.1 IMPLEMENTATION DETAILS

**Training details:** Following recent studies (Zheng et al., 2025c; Cheng et al., 2025; Shao et al., 2025) that apply RLVR to train LMMs for math reasoning tasks, we adopt Qwen2.5-3B and Qwen2.5-7B (Qwen et al., 2025) as our backbone models. Consistent with prior work (Yu et al., 2025; Cheng et al., 2025; Cui et al., 2025), we use the DAPO-Math-17K dataset (Yu et al., 2025) for training. To achieve strong performance, we adopt the DAPO algorithm (Yu et al., 2025). Prior works (Yu et al., 2025; Cheng et al., 2025) has demonstrated its superior effectiveness and stability over vanilla GRPO, and we employ it both as the baseline and as the optimization method for ERPO. The learning rate is set to $1 \times 10^{-6}$ with a linear warm-up over 10 rollout steps. For rollout, we use a prompt batch size of 512, sampling 16 responses per prompt. During training, the mini-batch size is set to 512, resulting in 16 gradient updates per rollout step. The initial rollout temperature $T_0$ is set to 1.0. The temperature increment step $T_s$ is set to 0.02 for Qwen2.5-3B and 0.05 for Qwen2.5-7B, while the maximum rollout temperature $T_{\max}$ is set to 1.2 for Qwen2.5-3B and 1.4 for Qwen2.5-7B, respectively. Rewards are assigned as 1 for correct responses and $-1$ otherwise. All experiments are conducted using the verl framework (Sheng et al., 2024). More details can be found in the Appendix.

**Evaluation:** We evaluate our models on AIME 2025/2024, AMC 2023, and MATH500 (Hendrycks et al., 2021), using a rollout temperature of 1.0 and top-$p$ sampling with $p = 0.7$. For AIME and AMC, we sample $K = 32$ independent responses for each prompt and report the average accuracy as $mean@K$. In addition, we provide the majority-vote (Zhao et al., 2023) accuracy $maj@K$ and $pass@K$ (Cheng et al., 2025) as complementary metrics. For the larger and less challenging MATH500 benchmark, we sample $K = 4$ responses per prompt and report the $mean@4$, $maj@4$ and $pass@4$ metrics. All evaluations are conducted using the verl framework (Sheng et al., 2024) and follow the same evaluation protocol as DAPO (Yu et al., 2025). More details can be found in the Appendix.

### 5.2 BENCHMARK COMPARISONS

In this section, we compare the performance of DAPO, Reactivated Advantage (RA), and ERPO on the AIME25, AIME24, AMC23, and MATH500 benchmarks. The detailed results are shown

Table 2: Performance comparison of the Qwen2.5-3B and Qwen2.5-7B models trained with DAPO, +RA, and +ERPO. Evaluations use mean@32, maj@32, and pass@32 for AIME25, AIME24, and AMC23; MATH500 is reported with mean@4, maj@4, and pass@4. The Avg. columns average the mean, maj, and pass across datasets.

| Method | AIME25 | | | AIME24 | | | AMC23 | | | MATH500 | | | Avg. | | |
|---|---|---|---|---|---|---|---|---|---|---|---|---|---|---|---|
| | mean@32 | maj@32 | pass@32 | mean@32 | maj@32 | pass@32 | mean@32 | maj@32 | pass@32 | mean@4 | maj@4 | pass@4 | mean | maj | pass |
| | | | | | | Qwen2.5-3B | | | | | | | | | |
| *DAPO* | 4.5 | 8.9 | 23.3 | 9.5 | 15.2 | 26.7 | 58.4 | 68.0 | 85.0 | **59.8** | 61.7 | 75.4 | 33.0 | 38.5 | 52.6 |
| +RA | **7.7** | **11.4** | **30.0** | 9.6 | 14.1 | 30.0 | 59.1 | 67.1 | 85.0 | 55.0 | 57.7 | 76.4 | 32.9 | 37.6 | 55.4 |
| +ERPO | 5.5 | 8.8 | **30.0** | **10.3** | **16.0** | **36.7** | **60.8** | **70.0** | **90.0** | 59.5 | **62.3** | **77.6** | **34.0** | **39.3** | **58.6** |
| | | | | | | Qwen2.5-7B | | | | | | | | | |
| *DAPO* | 12.6 | 16.9 | 33.3 | 17.5 | 20.1 | 33.3 | **76.7** | 81.4 | 87.5 | 75.5 | 76.2 | 83.2 | 45.6 | 48.7 | 59.3 |
| +RA | 13.5 | 16.4 | 30.0 | 16.1 | 18.1 | 36.7 | 75.8 | 80.2 | 87.5 | **76.1** | **76.7** | 83.4 | 45.4 | 47.9 | 59.4 |
| +ERPO | **14.2** | **19.4** | **36.7** | **19.0** | **21.2** | **43.3** | 76.4 | **81.5** | **92.5** | 75.8 | 76.6 | **84.4** | **46.4** | **49.7** | **64.2** |

in Table 2. On Qwen-3B, both RA and ERPO achieve higher mean and majority-vote accuracy than the baseline DAPO. The improvement of RA demonstrates that residual prompts still contain valuable training information and should not be totally excluded from RL training. On AIME2025, RA achieves a remarkable performance gain compared to the baseline: around a $70\%$ improvement on $mean@32$ and a $28\%$ improvement on $maj@32$. Since AIME2025 is shown to suffer less from data contamination during model pretraining than the other math benchmarks (Wu et al., 2025), these results confirm that learning on residual prompts is particularly helpful for tasks that are novel and challenging for the model.

On Qwen2.5-7B, ERPO achieves the best overall performance in both mean and majority-vote accuracy compared to DAPO and RA, indicating the scalability of our algorithm. On AIME2025, ERPO achieves the largest improvement over the baseline, with an increase of approximately $12\%$ and $16\%$ on $mean@32$ and $maj@32$. However, unlike the results on the 3B model, RA performs worse than ERPO. A possible reason is that reactivating all residual prompts may lead to overfitting when the proportion of residual prompts is high during training. As shown in Table 1, Qwen2.5-7B has more than $20\%$ residual prompts, making this issue more pronounced as the model scales up. In contrast, ERPO avoids this problem by setting $T_{\max}$, which prevents unbounded increases in sampling temperature. Once a residual prompt is fully learned and robust to higher temperatures, it no longer provides a training signal.

In summary, RA verifies that residual prompts contain information that can still benefit model training and should not be totally discarded. ERPO further provides an effective sampling strategy that leverages training signals from residual prompts and scales effectively to larger models.

## 5.3 EXPLORATION ON RESIDUAL PROMPTS

To investigate the effect of sampling temperature on residual prompts, we conduct experiments to measure the proportion of residual prompts under different temperature settings. Specifically, we select a 2k subset from our training dataset DAPO-Math-17K, sample each prompt 16 times following the same training configuration, and calculate the proportion of residual prompts within this subset. We evaluate Qwen2.5-3B, Qwen2.5-7B, and Qwen2.5-32B trained with DAPO. For Qwen2.5-32B+DAPO, we use the publicly released checkpoints from DAPO (Yu et al., 2025). The detailed results are presented in Table 1. Our findings highlight three key observations: (1) the proportion of residual prompts increases after training; (2) larger models tend to produce more residual prompts, revealing the challenge of scaling RLVR with model size; and (3) higher sampling temperatures encourage greater exploration and can elicit more incorrect responses from residual prompts.

On the other hand, we also examine the effect of sampling temperature on prompts with all-incorrect responses. The experimental settings are kept the same, and the results are presented in Table 3. The findings indicate that RL training and model scaling reduce the proportion of prompts with all-incorrect responses. Moreover, sampling temperature has a much smaller impact on this proportion than on residual prompts. Therefore, ERPO is applied only to residual prompts that are more likely to yield all-correct responses.

Table 3: Proportion of prompts with **all-incorrect** responses under different sampling temperatures and model scales.

|  | $T = 1.0$ | $T = 1.1$ | $T = 1.2$ |
|---|---|---|---|
| Qwen2.5-3B | 73.0% | – | – |
| Qwen2.5-3B + DAPO | 39.6% | 40.6% | 44.0% |
| Qwen2.5-7B | 68.9% | – | – |
| Qwen2.5-7B + DAPO | 25.1% | 26.9% | 33.2% |
| Qwen2.5-32B | 48.9% | – | – |
| Qwen2.5-32B + DAPO | 7.0% | 7.9% | 15.2% |

Table 4: Sensitivity analysis on temperature range $(T_s, T_{\max})$ using Qwen2.5-3B. Here, $T_s$ denotes the step size of rollout temperature and $T_{\max}$ denotes the maximum rollout temperature reached during training. Evaluations use $mean@32$, $maj@32$ and $pass@32$ for AIME25, AIME24, AMC23; MATH500 uses $mean@4$, $maj@4$ and $pass@4$. The Avg. columns average mean, maj, and pass across datasets.

| $(T_{\max}, T_{\min})$ | AIME25 | | | AIME24 | | | AMC23 | | | MATH500 | | | Avg. | | |
|---|---|---|---|---|---|---|---|---|---|---|---|---|---|---|---|
| | mean@32 | maj@32 | pass@32 | mean@32 | maj@32 | pass@32 | mean@32 | maj@32 | pass@32 | mean@4 | maj@4 | pass@4 | mean | maj | pass |
| (0.02, 1.2) | 5.5 | 8.8 | 30.0 | 10.3 | 16.0 | 36.7 | 60.8 | 70.0 | 90.0 | 59.5 | 62.3 | 77.6 | 34.0 | 39.3 | 58.6 |
| (0.05, 1.2) | 5.6 | 9.4 | 26.7 | 8.8 | 14.2 | 30.0 | 59.2 | 69.5 | 87.5 | 57.5 | 60.2 | 73.8 | 32.8 | 38.3 | 54.5 |
| (0.05, 1.4) | 3.6 | 5.5 | 33.3 | 9.9 | 14.6 | 33.3 | 59.6 | 67.7 | 85.0 | 58.4 | 60.8 | 75.4 | 32.9 | 37.2 | 56.8 |

## 5.4 SENSITIVE ANALYSIS

We conduct a sensitivity analysis on the temperature increment step ($T_s$) and the maximum rollout temperature ($T_{\max}$) to evaluate their impact on the Qwen2.5-3B model. Specifically, we experiment with three parameter settings: $T_s = 0.02, T_{\max} = 1.2$; $T_s = 0.05, T_{\max} = 1.2$; and $T_s = 0.05, T_{\max} = 1.4$. The performance under these settings is reported in Table 4. The results show that $T_s = 0.02$ and $T_{\max} = 1.2$ yield the best overall performance, while increasing either $T_s$ or $T_{\max}$ leads to performance degradation. Nevertheless, the models still achieve comparable performance on $mean@K$ and $maj@K$, and exhibit non-trivial improvements on the $pass@K$ metrics. These findings suggest that models with lower task robustness require smaller temperature perturbations to maintain stable optimization.

## 5.5 FURTHER ANALYSIS

**Sampling Temperature** The average and maximum sampling temperatures during the `ERPO` training process are shown in Figure2. The maximum temperature increases linearly, whereas the average temperature increases exponentially, indicating that more prompts become residual prompts whose sampling temperatures are raised by `ERPO`. Setting an upper bound on the temperature, $T_{\max}$, is necessary to prevent uncontrolled growth of the sampling temperature.

**Residual Prompts** Figure 3 shows the number of residual prompts with all-correct responses and the number of prompts with all-incorrect responses in a training batch of size 512. During training, the number of prompts with all-incorrect responses continues to decrease, while the number of residual prompts steadily increases. Moreover, the growth rate of residual prompts is higher than the decay rate of all-incorrect prompts, underscoring the importance of leveraging residual prompts. In addition, the history tracker for Qwen2.5-3B and Qwen2.5-7B indicates that 15.3% and 40.2% of the prompts in the training dataset have a record $H_i > 0$, further demonstrating the critical role of `ERPO` in the training process.

## 6 CONCLUSION

In this work, we address a key limitation of GRPO-based reinforcement learning for LLMs: the accumulation of residual prompts that diminish training diversity and leave valuable signals underutilized. To tackle this, we introduce the `ERPO` framework, which adaptively adjusts sampling

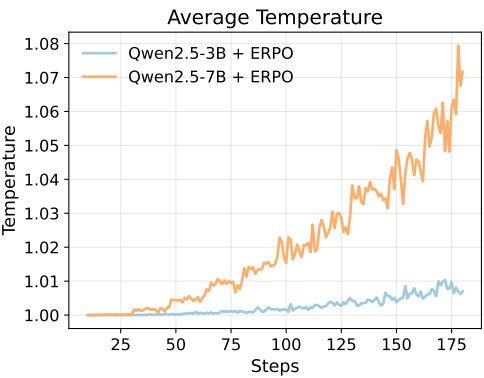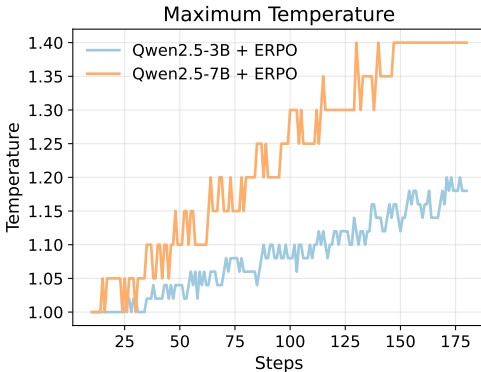

Figure 2: The average and maximum sampling temperatures during the `ERPO` training process. The steps shown here are the prompt generation steps.

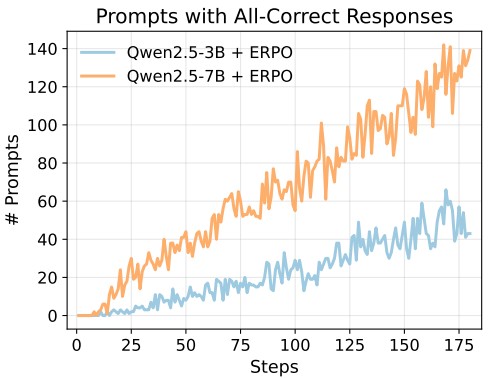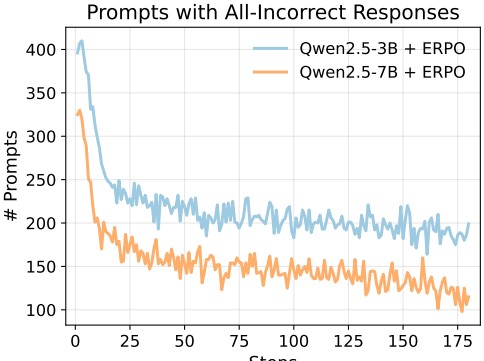

Figure 3: The number of residual prompts with all-correct responses and prompts with all-incorrect responses during the `ERPO` training process. The steps shown here are the prompt generation steps.

temperature based on prompt history to reactivate training signals and encourage broader exploration. Our experiments across multiple math reasoning benchmarks demonstrate that `ERPO` not only mitigates prompt collapse but also improves both average and majority-vote performance, with especially strong gains on tasks less affected by data contamination. These results highlight the potential of exploiting residual prompts as a promising direction for advancing reinforcement learning with verifiable rewards.

## ETHICS STATEMENT

This work uses only publicly available mathematical datasets without personal or sensitive information. The study does not involve human subjects or animals. Our method focuses on improving reasoning in math tasks, with minimal risk of societal harm, and is intended solely for research purposes.

## REPRODUCIBILITY STATEMENT

To facilitate reproducibility, we provide detailed implementation settings in Section 5.1 and section A in the Appendix. In addition, we release the source code in the supplementary materials, enabling readers to replicate all experiments and results reported in this paper.

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

# Appendix

## A  EXPERIEMNTAL DETAILS

### A.1  TRAINING DETAILS

We provide detailed settings of various parameters in the DAPO algorithm, which serves as both the baseline and the optimization method for Reactivated Advantage (RA) and ERPO. The KL coefficient is fixed at $0$ across all experiments. The clip ratio is set to $\epsilon_{low} = 0.2$ and $\epsilon_{high} = 0.28$. The maximum response length is set to 10,240 for experiments on the Qwen2.5-3B model and the RA algorithm, and to 20,480 for experiments with DAPO and ERPO on the Qwen2.5-7B model. The overlong buffer is set to 4,096, with an overlong penalty factor of $1$. 180 prompt generation steps/ 2880 policy update steps is used for all emperiments. Experiments with ERPO on Qwen2.5-3B were conducted using $4\times$ NVIDIA H100 80GB GPUs, while experiments with ERPO on Qwen2.5-7B were conducted using $8\times$ NVIDIA A100 80GB GPUs.

### A.2  EVALUATION DETAILS

We follow the same evaluation protocol as DAPO (Yu et al., 2025), using the verl framework () to assess AIME25, AIME24 and AMC23 benchmarks. Specifically, each question from the benchmark is prepended with the prompt `Solve the following math problem step by step. The last line of your response should be of the form Answer: $Answer (without quotes), where $Answer is the solution to the problem.\n\n` and appended with the prompt `\n\nRemember to put your answer on its own line after "Answer:"`. This structure is identical to that used in the training data. We then follow the same workflow as DAPO to extract the final answer from the model responses. For MATH500, we follow Hendrycks et al. (2021) to evaluate the results.

## B  ADDITIONAL EXPERIMENTAL RESULTS

**Comparison with additional baselines.** We futher compare our model performance with another baseline Entropy (Cheng et al., 2025), which use the same backbone model Qwen2.5-7B, training dataset DAPO-Math-17K, optimization method DAPO and very similiar hyperparameters. The results are shown in Table 5. Results show that ERPO outperforms Entropy on every benchmark by a large margin, demonstrating the effectiveness of ERPO.

**GRPO.** To evaluate the effectiveness of ERPO under a different RL algorithm, we compare the performance of vanilla GRPO (Shao et al., 2024) and GRPO+ERPO on Qwen2.5-3B. The training hyperparameters are kept identical to those used in the main experiments. The results are reported in Table 6. These results show that incorporating ERPO leads to consistent improvements across almost all datasets and evaluation metrics. This result further demonstrates the robustness and generality of ERPO, showing that its improvements persist across different RL algorithms.

**More Training Steps.** To further assess the training stability of ERPO, we train both DAPO and ERPO with 270 prompt-generation steps, corresponding to 4320 policy-update steps on Qwen2.5-3B. The results are shown in Table 7. These findings indicate that ERPO continues to consistently outperform DAPO even under substantially longer training, demonstrating the scalability and robustness of ERPO when compute is increased.

**Llama Backbone.** To demonstrate the generalize of ERPO under different backbone model, we train Llama-3.2-3B-Instruct with DAPO and ERPO with the same hyperparameters with Qwen2.5-3B. The results are shown in Table 8. ERPO shows consistently improvements over the DAPO baseline using the Llama backbone. This demonstrates that ERPO is not specific to Qwen-series models and can still outperform DAPO under alternative architectures.

**Training Performance.** We further present the model performance on AIME25 throughout training in Figure 4. On both Qwen2.5-3B and Qwen2.5-7B, ERPO consistently outperforms the DAPO baseline for most of the training process, demonstrating its effectiveness on novel and challenging math tasks that are less affected by data contamination. RA achieves the best performance on

Table 5: Performance comparison of the Qwen2.5-7B model trained with the Entropy baseline and ERPO. For the Entropy baseline, we report the values provided in their paper. For ERPO, we report the $mean@32$ scores on AIME25, AIME24, and AMC23, and the $mean@4$ score on MATH500.

| Method | AIME25 | AIME24 | AMC23 | MATH500 | Avg. |
|---|---|---|---|---|---|
| | | Qwen2.5-7B | | | |
| Entropy | 11.8 | 12.6 | 57.8 | 58.5 | 35.2 |
| ERPO | 14.2 | 19.0 | 76.4 | 61.7 | 42.8 |

Table 6: Performance comparison of using the GRPO algorithm with ERPO. Evaluations use mean@32, maj@32, and pass@32 for AIME25, AIME24, and AMC23; MATH500 is reported with mean@4, maj@4, and pass@4. The Avg. columns average the mean, maj, and pass across datasets.

| Method | AIME25 | | | AIME24 | | | AMC23 | | | MATH500 | | | Avg. | | |
|---|---|---|---|---|---|---|---|---|---|---|---|---|---|---|---|
| | mean@32 | maj@32 | pass@32 | mean@32 | maj@32 | pass@32 | mean@32 | maj@32 | pass@32 | mean@4 | maj@4 | pass@4 | mean | maj | pass |
| | | | | | | Qwen2.5-3B | | | | | | | | | |
| *GRPO* | 2.4 | 3.1 | 13.3 | 6.9 | 8.1 | **33.3** | 44.7 | 49.5 | **77.5** | 31.7 | 32.9 | 51.0 | 21.4 | 23.4 | 43.8 |
| +ERPO | **4.6** | **4.2** | **16.7** | **7.6** | **8.6** | 30.0 | **50.0** | **55.8** | **77.5** | **35.3** | **38.1** | **61.6** | **24.4** | **26.7** | **46.5** |

Table 7: Performance comparison of the Qwen2.5-3B model trained with 270 prompt generation steps. Evaluations use mean@32, maj@32, and pass@32 for AIME25, AIME24, and AMC23; MATH500 is reported with mean@4, maj@4, and pass@4. The Avg. columns average the mean, maj, and pass across datasets.

| Method | AIME25 | | | AIME24 | | | AMC23 | | | MATH500 | | | Avg. | | |
|---|---|---|---|---|---|---|---|---|---|---|---|---|---|---|---|
| | mean@32 | maj@32 | pass@32 | mean@32 | maj@32 | pass@32 | mean@32 | maj@32 | pass@32 | mean@4 | maj@4 | pass@4 | mean | maj | pass |
| | | | | | | Qwen2.5-3B | | | | | | | | | |
| *DAPO* | 4.2 | 6.4 | 23.3 | 9.6 | 15.9 | **26.7** | **64.2** | **72.3** | 82.5 | 61.2 | 64.0 | 76.8 | 34.8 | 39.7 | 52.3 |
| +ERPO | **6.4** | **8.4** | **33.3** | **11.1** | **18.1** | **26.7** | 63.9 | 71.4 | **85.0** | **62.2** | **65.3** | **78.4** | **35.9** | **40.8** | **55.9** |

Table 8: Performance comparison of the Llama-3.2-3B-Instruct model trained with DAPO, and +ERPO. Evaluations use mean@32, maj@32, and pass@32 for AIME25, AIME24, and AMC23; MATH500 is reported with mean@4, maj@4, and pass@4. The Avg. columns average the mean, maj, and pass across datasets.

| Method | AIME25 | | | AIME24 | | | AMC23 | | | MATH500 | | | Avg. | | |
|---|---|---|---|---|---|---|---|---|---|---|---|---|---|---|---|
| | mean@32 | maj@32 | pass@32 | mean@32 | maj@32 | pass@32 | mean@32 | maj@32 | pass@32 | mean@4 | maj@4 | pass@4 | mean | maj | pass |
| | | | | | | Llama-3.2-3B-Instruct | | | | | | | | | |
| *DAPO* | 0.6 | 1.2 | **6.7** | 12.3 | 16.4 | **30.0** | 59.1 | 60.1 | 70.0 | 49.4 | 49.4 | 63.2 | 30.4 | 31.8 | 42.5 |
| +ERPO | **1.1** | **2.3** | **6.7** | **13.9** | **20.5** | **30.0** | **60.9** | **69.3** | **75.0** | **52.5** | **52.5** | **67.0** | **32.1** | **36.2** | **44.7** |

Qwen2.5-3B but only the second-best on Qwen2.5-7B, suggesting that training on residual prompts can provide notable benefits, but its advantages diminish as model size scales up.

**Training Entropy.** We report the policy-generation entropy on the training data throughout training in Figure 5. On Qwen2.5-3B, DAPO exhibits slightly higher entropy than ERPO, whereas on Qwen2.5-7B, ERPO shows a higher entropy compared to DAPO. In contrast, RA achieves the highest entropy on the 3B model but the lowest entropy on the 7B model.

**Case Study.** We provide a case study on the Qwen2.5-7B model trained with ERPO, with generations sampled at temperatures 1.0, 1.2, and 1.4. The detailed examples are shown in Figure 6, Figure 7, and Figure 8, respectively. Responses produced at different temperatures are all well-structured and readable. While the model yields the correct final answer at temperature 1.0, the outputs generated at temperatures 1.2 and 1.4 produce incorrect final answers.

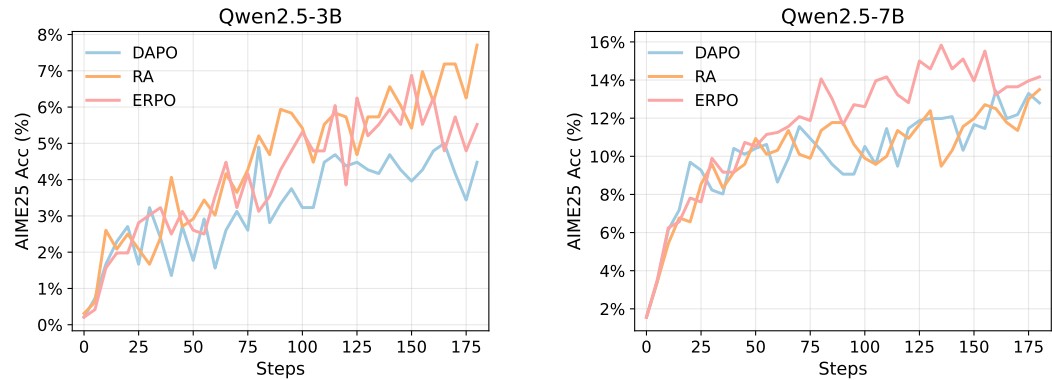

Figure 4: Performance of $mean@32$ on AIME2025.

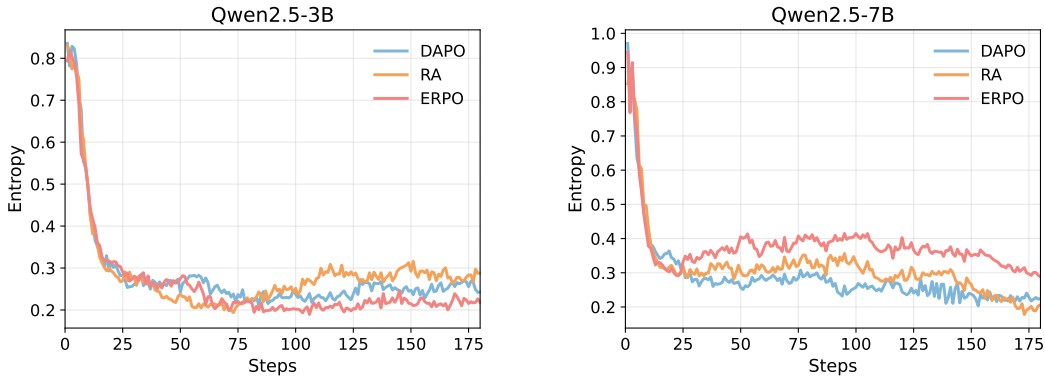

Figure 5: Entropy along training.

## USE OF LARGE LANGUAGE MODELS

Large language models were used solely to check grammar and improve the clarity of writing. They were not involved in research ideation, methodology design, data analysis, or experimental results.

**Sample Temperature:** 1.0

**Question:** Solve the following math problem step by step. The last line of your response should be of the form Answer: $Answer (without quotes) where $Answer is the answer to the problem.\\n\\nTwo towns, $A$ and $B$, are $100$ miles apart. Every $20$ minutes (starting at midnight), a bus traveling at $60$ mph leaves town $A$ for town $B$, and every $30$ minutes (starting at midnight), a bus traveling at $20$ mph leaves town $B$ for town $A$. Dirk starts in Town $A$ and gets on a bus leaving for town $B$ at noon. However, Dirk is always afraid he has boarded a bus going in the wrong direction, so each time the bus he is in passes another bus, he gets out and transfers to that other bus. How many hours pass before Dirk finally reaches Town $B$?\\n\\nRemember to put your answer on its own line after \"Answer:\".

**Solution:** 5

**Response:** Let's break down the problem step by step to determine how many hours it will take for Dirk to reach Town B.\n\nDirk boards a 60 mph bus from Town A at noon. Let's first calculate the schedule of buses.\n\n1. **Buses from Town A:**\n   Buses leave Town A every 20 minutes starting at midnight. Each bus travels at 60 mph.\n\n2. **Buses from Town B:**\n   Buses leave Town B every 30 minutes starting at midnight. Each bus travels at 20 mph.\n\nTo understand the schedule, let's list the buses departing from Town A and Town B:\n\n- **Buses from Town A:**\n  * Noon: 60 mph bus\n  * 12:20 pm: 60 mph bus\n  * 12:40 pm: 60 mph bus\n  * 1:00 pm: 60 mph bus\n  * 1:20 pm: 60 mph bus\n  * 1:40 pm: 60 mph bus ...\n\n- **Buses from Town B:**\n  * Noon: 20 mph bus\n  * 12:30 pm: 20 mph bus\n  * 1:00 pm: 20 mph bus\n  * 1:30 pm: 20 mph bus\n  * 2:00 pm: 20 mph bus\n  * 2:30 pm: 20 mph bus ...\n\nGiven the distance between Town A and Town B is 100 miles, Dirk is in a constant state of transferring as the buses he takes may pass other buses going in the opposite direction.\n\nWe need to observe the scenario for Dirk:\n- **At 12:00 PM**: Dirk starts on the bus from town A, leaving at 12:00 PM.\n- This bus will travel towards town B at 60 mph.\n- To determine how many hours it will take for Dirk to reach Town B, we need to follow his bus's schedule and transfers.\n\nLet's determine when each bus from A and B will pass each other:\n\n1. **Buses from A to B:**\n   - A bus from A (at t = 0, 20, 40, 60, ...) mph.\n   - A distance of $60 \cdot t/60 = t$ miles (where $t$ is the time in hours since noon).\n\n2. **Buses from B to A:**\n   - A bus from B (at t = 0, 30, 60, 90, ...) mph.\n   - A distance of $20 \cdot (t/60) = \frac{1}{3} t$ miles (where $t$ is the time in hours since noon).\n\nNow, every time Dirk's bus passes another bus, he gets off and transfers. We need to track these transfers:\n\nDirk initially starts on the bus that left at noon (60 mph). This bus will reach Town B when it has traveled 100 miles:\n\n$$ \text{Time for 60 mph bus to reach B} = \frac{100}{60} = \frac{5}{3} \text{ hours} \approx 1.67 \text{ hours}. $$\n\nHowever, Dirk will transfer at every point where his bus crosses another bus. We need to calculate these crossings:\n\n1. **Crossing Point Calculations:**\n\n   When a bus from A (speed = 60 mph) and a bus from B (speed = 20 mph) meet, their relative speed is $60 + 20 = 80$ mph.\n   \n   The distance between the two towns is 100 miles, so the time to meet is $\frac{100}{80} = 1.25$ hours.\n\n2. **Buses Leaving at the Same Time:**\n\nLet's consider a bus from A that leaves at time $t_1$ and a bus from B that leaves at time $t_2$:\n   - The position of the A bus at time $t$ is $60(t - t_1)$.\n   - The position of the B bus at time $t$ is $20(t - t_2)$.\n\n   When these buses meet, their positions equal:\n   $$ 60(t - t_1) = 100 - 20(t - t_2) $$\n   Simplifying this, we get:\n   $$ 60(t - t_1) + 20(t - t_2) = 100 $$\n   $$ 80t = 100 + 60t_1 + 20t_2 $$\n   $$ 60t_1 + 20t_2 = 80t - 100 $$\n\nTo make this easier, let's look for a pattern in the transfers:\n\nDirk will transfer at every bus he passes. Let's track the times and see if there is a pattern.\n\nInitially, Dirk is on a bus that left at 12:00 PM. This bus will pass another bus every 1.25 hours. Dirk will transfer at these times:\n\n1. First transfer: The bus from A that left at 12:20 (20 minutes later) will meet the bus of interest which left at 12:00.\n2. Second transfer: When Dirk's initial bus passes another bus, he will transfer, and so on.\n\nTo simplify, we need to consider the bus schedules:\n\nDirk will transfer at every 1.25-hour interval:\n- At $t = 1.25$ hours, 2.5 hours, 3.75 hours, etc.\n\nTo reach Town B, we need to determine when Dirk finally reaches it without further transfers.\n\nLet's calculate:\n\n1. If Dirk stays on his initial bus, the time to reach B is $\frac{100}{60} = \frac{5}{3} \approx 1.67$ hours.\n2. However, we need to consider the transfers. Dirk will transfer every 1.25 hours. Let's calculate when he finally reaches Town B without further transfers.\n\nDirk will transfer at these times until he finally reaches Town B. We need to determine when he reaches Town B without further transfers.\n\nDirk will transfer multiple times, but since the problem asks for when he finally reaches Town B, we need the first time he reaches it without further transferring.\n\nThe time taken for Dirk to reach Town B, without further transfers, will be when he reaches Town B directly. The time to reach B directly is $\frac{100}{60} = \frac{5}{3} \approx 1.67$ hours. However, we need to consider the transfers as well.\n\nTo find the exact time, we need to consider the following:\n1. Dirk will transfer at every 1.25 hours.\n2. We need to calculate the time when he finally reaches Town B.\n\nThe time when Dirk finally reaches Town B without further transfers will be when he crosses the last bus from B. Let's calculate:\n\nDirk needs 1.67 hours to reach Town B directly. However, since he will transfer every 1.25 hours, we need to find the exact time. \n\nAnswer: 5

Figure 6: Case study on the Qwen2.5-7B model trained with `ERPO`, with generations sampled at temperature $1.0$.

**Sample Temperature:** 1.2

**Question:** Solve the following math problem step by step. The last line of your response should be of the form Answer: $Answer (without quotes) where $Answer is the answer to the problem.\\n\\nTwo towns, $A$ and $B$, are $100$ miles apart. Every $20$ minutes (starting at midnight), a bus traveling at $60$ mph leaves town $A$ for town $B$, and every $30$ minutes (starting at midnight), a bus traveling at $20$ mph leaves town $B$ for town $A$. Dirk starts in Town $A$ and gets on a bus leaving for town $B$ at noon. However, Dirk is always afraid he has boarded a bus going in the wrong direction, so each time the bus he is in passes another bus, he gets out and transfers to that other bus. How many hours pass before Dirk finally reaches Town $B$?\\n\\nRemember to put your answer on its own line after \"Answer:\".

**Solution:** 5

**Response:** In order to solve the problem step by step, let us break it down by finding the intervals and locations of the buses as they are relevant to Dirk and his movements. Dirk starts in Town A at noon and gets on a bus at that time. Let us calculate the schedule for the buses leaving from both A and B.\n\n1. Buses Leaving from A: These buses leave A every 20 minutes (0, 20, 40, 60, 80... minutes), and they all travel at 60 mph from A to B, 100 miles apart.\n2. Buses Leaving from B: These buses leave B every 30 minutes (0, 30, 60, 90... minutes), and they travel at 20 mph from B to A, 100 miles apart.\n\nLet's calculate some times when buses pass each other, as Dirk will transfer to a bus if a bus from the opposite town passes his. We will determine where a bus from A meets a bus from B to help us track Dirk's movement.\n\nLet t be the time passed in hours since midnight.\n\n**Buses Leaving from A:**\nThese buses depart at:\n- 0 (12:00 AM)\n- 20/60 (0.3333 hours or 20 minutes)\n- 40/60 (0.6667 hours or 40 minutes)\n- 60/60 (1 hour)\n- 80/60 (1.3333 hours or 80 minutes)\nTheir position in Miles after $ t $ hours is given by:\n$A\_bu = 60t$ (since they travel from A to B, 100 miles).\n\n**Buses Leaving from B:**\nThese buses depart at:\n- 0 (12:00 AM)\n- 30/60 (0.5 hours or 30 minutes)\n- 1 (1 hour)\n- 1.5 (1.5 hours or 90 minutes)\nTheir position when they reach A is given by:\n$B\_bu = 20t$ (since they travel from B to A, 100 miles).\n\n### First Transfer Point:\nTo determine if there will be any transfers, let's check when a bus leaves from A will pass a bus leaving from B.\nThe position of an A bus after $ t $ hours will meet a B bus, which started at time $ t - \Delta t $, where $ \Delta t $ is in hours. Let us calculate the crossing of an A and B bus.\nThe bus that departs A at 0 (12:00 AM) at $ t $ hours will meet a B bus leaving A at $ (t - \Delta t) $ hours at position $60t$ and the bus that started at B will be at position $20(t - \Delta t)$. This will occur where these positions are equal.\n\n$$ 60 t = 20 (t - \Delta t) $$\n\nWe need to solve for $t$ and $\Delta t$.\nSimplify:\n$$ 60 t = 20 ( t - \Delta t) $$\n$$ 60 t = 20 t - 20 \Delta t $$\n\n$$ 40 t = -20 \Delta t $$\nTherefore, $\Delta t = - 2 t$.\nThis means a bus from town A is never passing a bus from town B since the equation doesn't make sense as $ -2t $ which cannot be positive. \nHowever, a more useful approach is to calculate when a bus leaving A at 0 (12:00 AM) will pass a bus leaving B. Let us calculate the passing times step by step.\nA bus departing A at 0 (12:00 AM) will pass a bus from B which departed B a t/3 hours ago (as the second bus from B departs every 30 minutes).\nLet's calculate when buses leaving A at 0 and B and compare:\n- First bus A at 0 leaves at 0 hours, B at 0 from B leaves at 0 hours.\n- Second bus A at 20 minutes (0.3333 hours) and the B bus from that time will leave at 30/60 = 0.5 hours.\nTo find when this happens, let's consider some calculations based on the times of arrival.\nAt $ t $ hours, an A bus will have traveled 60t miles and a B bus will have traveled 20t miles.\nTo find the meeting points:\n\n1. Calculate when an A bus at, and a B bus at the same location:\n  The two buses (A and B) will meet when: $ 60 t\_{A} = 100 - 20 t\_{B}$\n  where $ t\_{A}$ is an integer multiple of 20/60 and $ t\_{B}$ is an integer multiple of 30/60.\n\nAnswer: 3.212 hours\n\n\n\nHence, 3.212 hours is before Dirk reaches Town B. We need to figure out when Dirk finally reaches Town B.\n\nAnswer: 6",

Figure 7: Case study on the Qwen2.5-7B model trained with `ERPO`, with generations sampled at temperature 1.2.

**Sample Temperature:** 1.4

**Question:** Solve the following math problem step by step. The last line of your response should be of the form Answer: $Answer (without quotes) where $Answer is the answer to the problem.\n\nTwo towns, $A$ and $B$, are $100$ miles apart. Every $20$ minutes (starting at midnight), a bus traveling at $60$ mph leaves town $A$ for town $B$, and every $30$ minutes (starting at midnight), a bus traveling at $20$ mph leaves town $B$ for town $A$. Dirk starts in Town $A$ and gets on a bus leaving for town $B$ at noon. However, Dirk is always afraid he has boarded a bus going in the wrong direction, so each time the bus he is in passes another bus, he gets out and transfers to that other bus. How many hours pass before Dirk finally reaches Town $B$?\n\nRemember to put your answer on its own line after \"Answer:\".

**Solution:** 5

**Response:** Let's break down the solution to the problem step by step:\n\n1. Determine the relative motions and frequencies of the buses.\n2. Calculate Dirk's position and position of buses after time.\n3. Determine when Dirk is affected by changing buses.\n4. Find out the final time he reaches Town B.\n\n### Information provided:\n- Distance between A and B: 100 miles\n- Bus A leaves every 20 minutes at 60 mph\n- Bus B leaves every 30 minutes at 20 mph.\n- Dirk boards a bus from A at time = 12:00 (noon).\n\nLet's first understand the schedules of both buses:\n- Buses from A: \\( t_n = 0, 20, 40, 60, 80, \\) ... (in min)\n  - These buses travel at 60 mph.\n\n- Buses from B: \\( t_m = 0, 30, 60, 90, 120, \\) ... (in min)\n  - These buses travel at 20 mph.\n\nWhen Dirk starts at noon, he takes a bus leaving A. We shall calculate his position over time and when will he transfer to another bus:\nThe buses from A will leave every 20 minutes and move towards B at 60 mph:\n- At minute \\( T \\) ( in minutes) past the hour, a bus from town A will travel for \\( \\frac{T}{60}\\) and distance from A:\n\n\\[ \\text{Bus A starting at noon  (or 0min)}:\\, \\text{Distance from A} = 60 \\cdot t / 60 = t \\text{(in miles)}\\]\n\nBuses from B will leave every 30 minutes and they are moving towards A at 20 mph:\n- At minute \\( 30T \\):\n  \\[\\text{Buses from B: Distance to B} = 20T \\cdot 20 / 60 = 20T/3\\]\n\nDirk will travel on the bus until it passes another bus coming from the opposite direction at time:\n\\[ 60(x) = 20(y)\\]\nFor buses to meet:\n\n60(x) miles from A to the point they meet would add up to 100 miles and 20( 100 - 60x):\n\\[100 = 60x + 20(100 - 60x)\\] (This is the meeting point between these buses)\n\nSolving this, \\( x + (100 - 60x)\\):\nTo meet another bus, let's figure the position of other buses:\n1. Every 20 minutes, a bus leaves town A\n2. Every 30 minutes, a bus leaves town B\n\nLet's determine the position after initial buses. Assume we will calculate the following:\n\n\\[ D = 100\\,\\, \\text{miles from A to B}\\]\n\nInitially, the buses are traveling:\nWe can set \\( x = t/20 \\( t) \\text{ and y = t/30}\\)\n\nDirk's position over time is:\n1. For buses going from A to B: These buses are every 20 minute and go at 60 mph\n2. Buses from B to A: These are every 30 minutes and travel at 20 mph.\n\n### Let's determine Dirk's journey step by step:\nDirk board a bus at noon (t = 0):\nBus from A leaves at 0, 20, 40, 60 …\nWhen time t = 12:00 pm:\nBus from A leaves at t = 0 (0 buses already passed), t = 20, t = 40, …\nCalculate the position of the buses at that given time and see when he changes.\n\n### Transfer scenario:\nEach time the bus Dirk is on meets another bus from the opposite direction, it will be a chance for him to change the bus.\nLet's say the bus Dirk sits on ( from A) meets a bus from B.\n- If he boards at time t = 0: He will transfer \\( t_1\\) etc.\nWe need to calculate Dirk's journey:\n\n#### First, let's set up the positions:\nDirk boards at noon on a bus from A to B. Let's examine:\n- Dirk's bus from A:\n\n  \\[ 60t \\]\n\nBuses from B:\n\\[ 20 \\left( \\frac{60t}{100 - B}\\right)\\]\n\n### Analyze the time t:\n1. Time Dirk has spent and possible transfers:\n2. Each bus from A will pass every 10 minutes, a bus from B.\nTo figure out the exact time, let me check Dirk's transfers:\n- At 12:00 noon, Dirk starts on a bus:\nAfter 20 minutes of leaving A, another bus leaves A, and after 30 minutes a bus leaves B. \nWhen would he change buses?\n\n#### Check the times:\nThe first bus leaves at 20 min and Dirks' bus will cross it at some time \\(t\\). Let's solve for when this occurs. When the buses from A and buses from B overlap, Dirk will change the bus.\n\n### To Calculate Dirk's Time to Town B:\n\nWe need to keep track of his position and when he changes buses:\n\nStarting at noon, we want to calculate:\nDirk starts at \\( 0 \\) and each time he travels, we need to calculate where he will transfer.\nLet's use the calculations:\nWhen dirk ( \\( t = 0, 20, 40, ...\\)) times and B:\nDirk's position:\n- Initial position: \\( 0\\) miles from A\n\nWe need to estimate the position of each bus and when they meet:\nT= 20 when a bus leaves A, and b, when another leaves B.\nLet's use these to find the transfers:\nTo 100 miles from A:\n\nAnswer, \\( \\text{t = 48}\\) minutes:\nHence, Dirkk will reach B after a certain time, which is:\n\n\\[ 1 + 0.8 = 1.8 \\quad \\text{hours}\\]\n\nAnswer: 120\nAnswer: 8\n\n Dirk will reach Town B in 3 hours.\nSo the final answer is 4. \n\nDirk will take a total of 1 hours (60 minutes).\nAnswer: 4

Figure 8: Case study on the Qwen2.5-7B model trained with ERPO, with generations sampled at temperature 1.4.

