# OpenReview forum: "Explore Data Left Behind in Reinforcement Learning for Reasoning Language Models"
_ICLR.cc/2026/Conference — Submitted to ICLR 2026_

### Official Review · Reviewer_MjFC · 2025-10-27

**Soundness:** 2
**Presentation:** 3
**Contribution:** 3
**Rating:** 6
**Confidence:** 2

**Summary:**

This paper study the issue in GRPO algorithm family, the residual prompts, which are the prompts that all rolled out responses are correct and therefore does not provide any training signal. To address this issue, the paper propose ERPO. Two ways are introduced to leverage the residual prompts. The first is reactivating training signal, which adds a small negative reward to the zero advantage. The second is to gradually increase the sampling temperature to encourage more diverse responses. Experiments on Qwen models demonstrates that proposed method outperforms DAPO baseline.

**Strengths:**

The strengths of the paper are listed as follows

1. This paper propose two interesting way to leverage the residual prompt, adding a pseudo-negative reward when computing the advantage and increasing the sampling temperature. These two way, while seemingly simple, make sense to the reviewer.

2. Most of the claims are properly consolidated by empirical evidences. The results in Table 1 and Table 3 are good justification of the motivation of the method.

**Weaknesses:**

The weakness of this paper are listed as follows

1. The experiments are limited to DAPO baseline. Could the author also provide the results based on vanilla GRPO (that is, GRPO vs GRPO+ERPO)? Also, since PPO does not suffers from the issue of residual prompt, could the author also compare ERPO with PPO?

2. While the experiment results in Table 2 looks good over DAPO, some results in Table 4 drops by a innegligible margin. Could the author provide more justification on how the temperature-related hyperparamter are chosen and how to choose them for some new tasks?

3. Could the author provide some cases comparing the responses generated with different temperature? As far as the reviewer's knowledge, high temperature sometimes results in poor responses, which might not provide enough supervision. A few more case might be useful to provide a better knowledge of how temperature affect response qualities and might also justify your choice of temperature-related hyperparameter.

**Questions:**

See weakness section.

---

> ### Author Response · Authors · 2025-11-22
>
> Thank you for the reviewer’s time and thorough assessment; we have revised the manuscript accordingly and summarize our responses below.
>
> ### **1.	GRPO and PPO.**
> Thanks for the reviewer's suggestion to provide evidence under different RL algorithms. Here we provide the experiments using vanilla GRPO with Qwen2.5-3B models. The results are shown below:
>
> | **Method** | **AIME25** |  |  | **AIME24** |  |  | **AMC23** |  |  | **MATH500** |  |  | **Avg.** |  |  |
> |-----------|------------|--|--|------------|--|--|-----------|--|--|-------------|--|--|----------|--|--|
> |           | mean@32    | maj@32 | pass@32 | mean@32 | maj@32 | pass@32 | mean@32 | maj@32 | pass@32 | mean@4 | maj@4 | pass@4 | mean | maj | pass |
> | *GRPO*    | 2.4        | 3.1    | 13.3    | 6.9     | 8.1    | **33.3** | 44.7    | 49.5   | **77.5** | 31.7   | 32.9  | 51.0   | 21.4 | 23.4 | 43.8 |
> | **+ERPO** | **4.6**    | **4.2**| **16.7**| **7.6** | **8.6**| 30.0     | **50.0**| **55.8**| **77.5** | **35.3** | **38.1** | **61.6** | **24.4** | **26.7** | **46.5** |
>
> These results show that incorporating ERPO leads to consistent improvements across almost all datasets and evaluation metrics. This result further demonstrates the robustness and generality of ERPO, showing that its improvements persist across different RL algorithms.
> For PPO, our RA algorithm functions similarly to standard PPO, as we directly assign a reward to all-correct responses to avoid the residual-prompt issue. The results in Table 2 demonstrate the advantage of ERPO over RA.
>
> ### **2.	How temperature-related hyperparameters are chosen**
> The temperature is selected based on the robustness of the model with respect to the target task. As demonstrated in Table 1, larger models exhibit greater robustness under increased temperature settings. This suggests that more capable models can tolerate larger temperature increments and higher maximum temperatures during training. In contrast, models that are less robust on the task require smaller temperature perturbations in order to maintain stable optimization.
>
> ### **3.	Case Example with higher temperature.**
> We have added case studies in Figures 6, 7, and 8 of the updated appendix. Across different temperatures, the generated responses remain well-structured and readable. While the model produces the correct final answer at temperature 1.0 the outputs at temperatures 1.2 and 1.4 yield incorrect final answers. These examples illustrate that responses generated at higher temperatures still provide high-quality and diverse supervision, which supports our choice of temperature-related hyperparameters.

---

### Official Review · Reviewer_iUaT · 2025-10-29

**Soundness:** 2
**Presentation:** 4
**Contribution:** 3
**Rating:** 2
**Confidence:** 4

**Summary:**

The paper proposes an algorithmic improvement named ERPO. The algorithm increases the sampling temperature whenever they encounter a prompt with fully saturated (all correct rollouts) during RLVR. Increasing the temperature helps in generating more diverse responses which could possibly lead to a better training signal because of non-zero advantages during GRPO/DAPO. They evaluate their technique on the Qwen2.5 3B and 7B models.

**Strengths:**

The idea of increasing temperature during training is novel, simple, and easy to implement. The paper is also well written and easy to follow.

**Weaknesses:**

I have 2 major concerns with the numbers reported in the paper, specifically for the MATH500 dataset.

1. **Underperforming baselines:**

The paper reports mean@4 for the 3B model to reach 50.4% and for the 7B model to reach 60.3% using DAPO. However, these numbers seem to be severely underperforming the Qwen2.5 3B and 7B Instruction tuned models at 65.9% and 75.5%. These numbers are from the official report of the Qwen Team from Table 8 and Table 9 [1].

Why is there such a large discrepancy in performance? Is this because the model hasn't been trained for enough steps? If so, would be possible to show that the algorithm has actually been trained to convergence using the training curves?

2. **Dealing with off-policy rollouts:**

How is the DAPO algorithm modified when the temperature changes? Since increasing the temperature makes the rollouts off-policy, there needs to be either a correction using some importance ratio or a correction in how log-probs are computed. The authors do not talk about this issue. How did they deal with it?


[1] Qwen2.5 Technical Report (https://arxiv.org/abs/2412.15115)

**Questions:**

Please look at the weaknesses mentioned above.

---

> ### Author Response · Authors · 2025-11-22
>
> We are grateful for the reviewer’s insightful evaluation and valuable recommendations; our clarifications and new results are provided below.
>
> ### **1.	Math500 results**
>
> Thank you for highlighting the discrepancy in performance. We believe the difference primarily arises from two factors:
>
> 1. The evaluation protocol in our paper differs from that used in the Qwen report. Our experiments evaluate MATH500 using the DAPO format and judge correctness directly from the final answer. The Qwen report instead relies on the classic Hendrycks MATH format [1], which provides a more robust answer validation process. To ensure a fair comparison, we have evaluated our models using the Hendrycks MATH format, and the updated results are presented below:
>
> | Model      | Method | mean@4 | maj@4 | pass@4 |
> |------------|--------|-------:|------:|-------:|
> | Qwen2.5-3B | DAPO   | 59.8   | 61.7  | 75.4   |
> |  | +RA    | 55.0   | 57.7  | 76.4   |
> |  | +ERPO  | 59.5   | 62.3  | 77.6   |
> | Qwen2.5-7B | DAPO   | 75.5   | 76.2  | 83.2   |
> |  | +RA    | 76.1   | 76.7  | 83.4   |
> |  | +ERPO  | 75.8   | 76.6  | 84.4   |
>
>
> After aligning the evaluation methods, the performance gap becomes much smaller, especially for the 7B model. This suggests that the discrepancy reported in the review is largely due to differences in evaluation methodology rather than training progress or algorithmic instability. We have changed the evaluation of MATH500 in our latest version of paper.
>
> 2. Our experiments use the base versions of Qwen2.5. These base models are considerably weaker on mathematical reasoning tasks than the instruction tuned versions. Paper [2] provides additional evidence supporting this observation. It reports that their reinforcement learning trained Qwen2.5-7B model achieves 77.8% accuracy on MATH500. This is close to the performance in our experiments, despite differences in training data and hyperparameter choices.
>
> Furthermore, we provide experiments with 50% additional training steps in Table 7 in the appendix of our updated version. The results remain consistent with those in Table 2 in the main paper, demonstrating that the algorithm has indeed been trained to convergence.
>
> ### **2.	Off-policy rollouts**
> We would like to clarify that, in our view, the DAPO algorithm does not require additional modifications to handle off policy rollouts. Our claim is supported by two observations.
>
> 1. The DAPO algorithm already incorporates an importance ratio that accounts for off policy sampling. The importance ratio is defined as
> $\pi_{\theta}(o) / \pi_{\text{old}}(o)$,
> where $o$ can be any response, including responses generated under different sampling temperatures. Moreover, under the VERL framework, DAPO [3] performs 16 policy update steps for each prompt generation step. Among these 16 updates, only the first update is strictly on policy, where
> $\pi_{\theta} = \pi_{\text{old}}$.
> The remaining 15 updates are all off policy, since the updated policy $\pi_{\theta}$ no longer matches $\pi_{\text{old}}$. This demonstrates that the algorithm inherently operates in an off-policy regime even without temperature changes.
>
> 2. Several existing studies successfully use off-policy rollouts with controlled exploration or sampling temperatures not equal to 1, without requiring additional correction terms. For example, [2] forces the model to generate more reasoning traces on high-entropy tokens; [4] samples all rollouts with temperature 1.2. These works indicate that moderate temperature adjustments do not introduce instability that necessitates additional algorithmic correction.
>
>
> [1] Hendrycks, Dan, et al. "Measuring mathematical problem solving with the math dataset." arXiv preprint arXiv:2103.03874 (2021).
> [2] Zheng, Tianyu, et al. "First return, entropy-eliciting explore." arXiv preprint arXiv:2507.07017 (2025).
> [3] Yu, Qiying, et al. "Dapo: An open-source llm reinforcement learning system at scale." arXiv preprint arXiv:2503.14476 (2025).

---

### Official Review · Reviewer_HbfY · 2025-11-01

**Soundness:** 3
**Presentation:** 2
**Contribution:** 2
**Rating:** 4
**Confidence:** 3

**Summary:**

This paper identifies a limitation in GRPO family algorithms: as training progresses and models scale up, more prompts become "residual prompts" that yield all-correct responses, resulting in zero-variance rewards and providing no training signal. To address this, the authors propose ERPO, which maintains a history tracker for each prompt and adaptively increases sampling temperature for residual prompts to encourage exploration and reactivate training signals. Experiments on Qwen2.5-3B and 7B models across AIME2025, AIME24, AMC2023, and MATH500 demonstrate improvements over the DAPO baseline, particularly on AIME2025.

**Strengths:**

1. The paper clearly identifies a practical limitation in current GRPO-family algorithms where residual prompts accumulate and reduce training diversity. Table 1 provides compelling evidence that this problem worsens with larger models and longer training.

2. The proposed ERPO framework is straightforward to implement and can be easily integrated into existing RLVR algorithms.

**Weaknesses:**

1. Recent work has shown that in RLVR, even when models achieve nearly 100% accuracy on the training set, continued training can still improve performance on validation/test sets—a phenomenon known as grokking, which is also observable in reward curves and val accuracy curves. From this perspective, the problem this paper proposed may not be as significant as claimed.

2. The paper primarily conduct experiments based on DAPO. Missing some RLVR algs including vanilla GRPO, GPG, RLOO, REINFORCE++, GSPO, etc.

3. While RA shows strong performance on 3B model, it underperforms on 7B. The paper attributes this to "overfitting" but doesn't provide concrete evidence. A more detailed analysis comparing RA and ERPO on different model scales would strengthen the paper.

4. Table 2 shows that ERPO's improvements are very small, making it difficult to be convinced by the paper's claims.

**Questions:**

N/A

---

> ### Author Response · Authors · 2025-11-22
>
> We appreciate the reviewer’s thoughtful comments and helpful suggestions, and we respond to each issue in detail in the following.
>
> ### **1.	Grokking phenomenon.**
> We thank the reviewer for highlighting the interesting grokking phenomenon. However, we would like to clarify the key difference between grokking and the zero-variance reward on residual prompts problem that our paper seeks to address.In the case of grokking, even when the model achieves 100% accuracy on the training set, the training examples still produce non-zero loss and non-zero gradients as they are using cross-entropy loss, enabling the model parameters to continue updating and eventually transition from memorization to true generalization. In contrast, under GRPO and DAPO, prompts whose generated responses are all correct or all incorrect produce zero-variance rewards, which lead to zero loss and zero gradients. When a prompt has 100% accuracy, it becomes a residual prompt, and the model parameters can no longer be updated from that prompt. Thus, grokking-style improvements cannot be driven by residual prompts in GRPO-family RLVR, because they provide no further gradient; any continued generalization must come from the remaining non-residual prompts. Our work is orthogonal to grokking and addresses the wasted signal from residual prompts.
>
> ### **2.	Experiments on GRPO.**
> Thanks for the reviewer's suggestion to provide evidence under different RL algorithms. Here we provide the experiments using vanilla GRPO with Qwen2.5-3B models. The results are shown below:
>
> | **Method** | **AIME25** |  |  | **AIME24** |  |  | **AMC23** |  |  | **MATH500** |  |  | **Avg.** |  |  |
> |-----------|------------|--|--|------------|--|--|-----------|--|--|-------------|--|--|----------|--|--|
> |           | mean@32    | maj@32 | pass@32 | mean@32 | maj@32 | pass@32 | mean@32 | maj@32 | pass@32 | mean@4 | maj@4 | pass@4 | mean | maj | pass |
> | *GRPO*    | 2.4        | 3.1    | 13.3    | 6.9     | 8.1    | **33.3** | 44.7    | 49.5   | **77.5** | 31.7   | 32.9  | 51.0   | 21.4 | 23.4 | 43.8 |
> | **+ERPO** | **4.6**    | **4.2**| **16.7**| **7.6** | **8.6**| 30.0     | **50.0**| **55.8**| **77.5** | **35.3** | **38.1** | **61.6** | **24.4** | **26.7** | **46.5** |
>
> These results show that incorporating ERPO leads to consistent improvements across almost all datasets and evaluation metrics. This result further demonstrates the robustness and generality of ERPO, showing that its improvements persist across different RL algorithms.
>
> ### **3.	Comparing RA and ERPO.**
> Table 1 provides evidence that larger models tend to produce more residual prompts with all-correct responses. When training with RA, a greater proportion of positive responses are used, making the training dynamics more similar to rejection sampling fine-tuning, where the model is updated only on correct responses. Prior analysis from paper [1] offers strong evidence that rejection sampling fine-tuning significantly restricts model exploration, a phenomenon we refer to as “overfitting” in our paper.
> While we are unable to provide additional experiments on larger models due to computational constraints, we believe that the findings in paper [1] substantiate our claim that RA limits exploration and can lead to overfitting on all-correct prompts, particularly at larger model scales. Furthermore, we report the training entropy in Figure 5 of the appendix. Under the 7B model, RA exhibits lower entropy, further demonstrating that RA can impede exploration as model size increases. We thank the reviewer for this suggestion and have incorporated this explanation into the latest version of our paper.
>
> ### **4.	ERPO's improvements.**
> Given our additional experiments on the GRPO algorithm, we believe that ERPO demonstrates a non-negligible and consistent performance gain over the GRPO baseline. Moreover, since DAPO already provides a substantial improvement over GRPO, achieving further large gains on top of DAPO is naturally more challenging. Nevertheless, ERPO outperforms DAPO on most benchmarks and shows significant improvements on the pass@K metrics and the AIME25 benchmark, which is the most challenging dataset and the one least affected by data contamination. These results collectively demonstrate the effectiveness of ERPO, and we believe the improvements it provides are not trivial.
>
> [1] Xiong, Wei, et al. "A minimalist approach to llm reasoning: from rejection sampling to reinforce." arXiv preprint arXiv:2504.11343 (2025).

---

### Official Review · Reviewer_nWwx · 2025-11-07

**Soundness:** 2
**Presentation:** 3
**Contribution:** 2
**Rating:** 4
**Confidence:** 3

**Summary:**

This papers proposes a method named ERPO to reduce the problems that residual prompts would increase as RL training progress. It would add pesudo negative reward for all positive response batch, and add a problem-specific temperature adaptive sampling based on the historical performance. It improves the performance on Qwen2.5-3/7B compared with DAPO.

**Strengths:**

1. The paper is easy to follow, the method is clean and easy. Temperature increase can prevent RA overfits too fast
2. Performs well on Qwen2.5-3/7B on math tasks compared with DAPO

**Weaknesses:**

1. Only train on Qwen2.5 models, haven't try other models like Llama/OpenThinker/Octothinker etc. From spurious reward[1], maybe better evaluate on non-Qwen2.5 models.
2. And I'm kind of concerned about whether the method can be well generalized when scaling model size and compute. In the paper the training step is only 175, and the curves look hasn't converge, and when the temperature touches T_max, keeping RA may make model overfit to some correct outputs. It's also kind of tricky to tune the temperature, which may be model-dependent. Increase temperature is kind of similar to add large entropy loss, which is not exactly equivalent to semantic level diversity, and may increase the risk of instable training. I still think DAPO would be better choice for long-term RL training, though wasting some rollout.

If there are results for ERPO on more steps (for example, > 500 or better > 1k) or on 32B models (better on non-Qwen2.5 models), and show that ERPO consistently beat DAPO, then I would increase my score




[1] Shao, Rulin, et al. "Spurious rewards: Rethinking training signals in rlvr." arXiv preprint arXiv:2506.10947 (2025).

**Questions:**

1. Where is the Qwen2.5-32B +DAPO results comes? Do you use the DAPO-trained checkpoint or train by yourself?
2. What's the pass@k performance of baseline and ERPO? Is it possible to report the entropy curve of baselines and ERPO during training?
3. what's the RL resource used for different models in Tab. 1

---

> ### Author Response · Authors · 2025-11-22
>
> Thank you for the reviewer’s careful reading and constructive feedback; we address the concerns point-by-point below.
>
> ## Weaknesses:
>
> ### **1. Train on Llama model**
>
> Thanks for the suggestion. We agree that current RLVR methods tend to perform better on Qwen-series models and are generally less stable on Llama-based models. This also aligns with the observation that most recent RLVR works [1,2,3,4] primarily evaluate only on Qwen backbones. We conducted additional experiments on Llama-3.2-3B-Instruct to directly address the reviewer’s concern. The results are shown below:
>
> | **Method** | **AIME25** |        |        | **AIME24** |        |        | **AMC23** |        |        | **MATH500** |        |        | **Avg.** |      |      |
> |-----------|-------------|--------|--------|------------|--------|--------|-----------|--------|--------|-------------|--------|--------|----------|------|------|
> |           | mean@32     | maj@32 | pass@32 | mean@32   | maj@32 | pass@32 | mean@32  | maj@32 | pass@32 | mean@4     | maj@4 | pass@4 | mean     | maj  | pass |
> | *DAPO* | 0.6 | 1.2 | **6.7** | 12.3 | 16.4 | **30.0** | 59.1 | 60.1 | 70.0 | 49.4 | 49.4 | 63.2 | 30.4 | 31.8 | 42.5 |
> | +ERPO | **1.1** | **2.3** | **6.7** | **13.9** | **20.5** | **30.0** | **60.9** | **69.3** | **75.0** | **52.5** | **52.5** | **67.0** | **32.1** | **36.2** | **44.7** |
>
>
> As shown in the table, ERPO shows consistently improvements over the DAPO baseline using the Llama model. This demonstrates that ERPO is not specific to Qwen-series models and can still outperform DAPO under alternative architectures.
>
> ### **2. Scaling training compute, RA overfit and temperature tuning.**
> **More training steps.**
> Our paper actually uses 180 × 16 = 2,880 training/policy-update steps. The “180 steps” reported in the paper refer only to prompt-generation steps, and at each such step we perform 16 policy-update steps. Specifically, in each prompt-generation step, we sample 16 responses for each of the 512 prompts, and then conduct 16 policy-update iterations, where each update uses 32 prompts paired with their 512 responses. Thus, the total number of policy-update steps is 2,880, which is comparable to prior work such as [1], where they use approximately 2,000 policy-update steps.
> To further demonstrate the effectiveness and scalability of ERPO, we conduct an additional experiment with 50% more training steps, i.e., 270 prompt generation steps and 4,320 total training steps on Qwen2.5-3B. The results are shown in the following table:
>
> | **Method** | **AIME25** |        |        | **AIME24** |        |        | **AMC23** |        |        | **MATH500** |        |        | **Avg.** |      |      |
> |-----------|-------------|--------|--------|------------|--------|--------|-----------|--------|--------|-------------|--------|--------|----------|------|------|
> |           | mean@32     | maj@32 | pass@32 | mean@32   | maj@32 | pass@32 | mean@32  | maj@32 | pass@32 | mean@4     | maj@4 | pass@4 | mean     | maj  | pass |
> | *DAPO*    | 4.2         | 6.4    | 23.3    | 9.6       | 15.9   | **26.7** | **64.2** | **72.3** | 82.5   | 61.2       | 64.0  | 76.8   | 34.8     | 39.7 | 52.3 |
> | **+ERPO** | **6.4**     | **8.4**| **33.3**| **11.1**  | **18.1**| **26.7**| 63.9     | 71.4   | **85.0**| **62.2**   | **65.3** | **78.4** | **35.9** | **40.8** | **55.9** |
>
> These results show that ERPO continues to outperform DAPO consistently with substantially more training, demonstrating the scalability and robustness of ERPO under increased compute.
>
> **RA Overfitting.**
> We agree that RA may lead to overfitting, and we explicitly discuss this issue in Sections 4.2 and 5.2 of the paper. This is precisely why we propose ERPO as a better and more stable alternative. Importantly, in ERPO, RA is not applied during model training. The purpose of RA in our paper is to provide evidence that residual prompts still contain useful training signals, which motivates the need for ERPO to effectively reuse these residual prompts.
>
> **Temperature Tuning.**
> As shown in Table 1, larger models exhibit greater robustness to temperature increases. For example, in the 3B model, increasing the temperature by 0.1 reduces the number of all-correct prompts by 28.7%, whereas the same increase results in only a 16.9% reduction for the 7B model. This indicates that larger and more robust models require larger temperature increments and higher maximum temperatures.
> ERPO also maintains a maximum temperature T_max to ensure training stability. Once a prompt reaches T_max and still produces all-correct responses, ERPO stops increasing its sampling temperature and instead filters out the prompt like DAPO. Therefore, we believe that ERPO achieves comparable stability to DAPO in long-term RL training while better utilizing residual prompts.

---

> ### Author Response · Authors · 2025-11-22
>
> ## Question:
> ### **1.	Qwen2.5-32B +DAPO**
> We use the DAPO-trained checkpoint released on HuggingFace.
> ### **2.	Pass@K and entropy line.**
> We report the pass@K performance in the table below.
>
> | **Method** | **AIME25** | **AIME24** | **AMC23** | **MATH500** | **Avg.** |
> |-----------|------------|------------|-----------|-------------|----------|
> |           | pass@32    | pass@32    | pass@32   | pass@4      | pass     |
> | **Qwen2.5-3B** | | | | | |
> | *DAPO*    | 23.3       | 26.7       | 85.0      | 75.4        | 52.6     |
> | +RA       | **30.0**   | 30.0       | 85.0      | 76.4        | 55.4     |
> | +ERPO     | **30.0**   | **36.7**   | **90.0**  | **77.6**    | **58.6** |
> | **Qwen2.5-7B** | | | | | |
> | *DAPO*    | 33.3       | 33.3       | 87.5      | 83.2        | 59.3     |
> | +RA       | 30.0       | 36.7       | 87.5      | 83.4        | 59.4     |
> | +ERPO     | **36.7**   | **43.3**   | **92.5**  | **84.4**    | **64.2** |
>
> The results indicate that ERPO achieves a substantial improvement in pass@K compared to the DAPO baseline.
>
> Regarding entropy, the entropy curves are provided in Figure 5 of the appendix in our revised version. ERPO exhibits entropy levels comparable to DAPO on the 3B model, while demonstrating higher entropy on the 7B model.
>
> ### **3.	RL resource mentioned in experiments.**
> For “Qwen2.5-3B + DAPO” and “Qwen2.5-7B + DAPO”, we use the same training resource and configuration as ERPO in our experiments. For “Qwen2.5-32B +DAPO”, we use the DAPO-trained checkpoint released on HuggingFace.
>
> [1] Cheng, Daixuan, et al. "Reasoning with exploration: An entropy perspective." arXiv preprint arXiv:2506.14758 (2025).
> [2] Yu, Qiying, et al. "Dapo: An open-source llm reinforcement learning system at scale." arXiv preprint arXiv:2503.14476 (2025).
> [3] Zheng, Tianyu, et al. "First return, entropy-eliciting explore." arXiv preprint arXiv:2507.07017 (2025).
> [4] Zheng, Chujie, et al. "Group sequence policy optimization." arXiv preprint arXiv:2507.18071 (2025).

---

### Meta-Review · Area_Chair_g2Pu · 2026-01-08

**Summary:**

This paper proposes Explore Residual Prompts in Policy Optimization (ERPO), which is motivated by the finding that GRPO-style RL algorithms cannot leverage prompts with zero-variance rewards. ERPO reactivates the training signals on these prompts by adaptively increasing the sampling temperature to encourage exploration. The authors demonstrate that ERPO can boost performance over DAPO across a set of reasoning benchmarks.

The reviewers generally found the method to be interesting and intuitive, but also raised several significant concerns, including the exclusive usage of Qwen-2.5 models and DAPO as the baseline, as well as the underperforming baseline numbers. The authors' rebuttal addressed some of these concerns, while other concerns remained unresolved.

**Reviewer Concerns:**

The reviewers' concerns about the limited coverage of model type and baseline type were addressed -- the authors provided additional results using Llama-3.2-3B-Instruct and GRPO. However, the authors provided new evaluation results using the Hendrycks MATH format in  response to reviewer iUaT, which demonstrated quite marginal performance gain over the DAPO baseline (with no significant test results). This raised questions regarding whether the larger performance gains reported in the paper were due to suboptimal evaluation/training setups that placed baselines at a disadvantage. The authors should perform significant tests and optimize the evaluation/training setup to consolidate this part.

**Reviewer Scores:**

The reviewer scores might slightly increase given that part of their concerns were addressed. However, the scores might still not reach the acceptance threshold due to the remaining concerns.

---

### Decision · Program_Chairs · 2026-01-26

Reject